# Developing Long-Term Sustainable Collaborations between Welfare Providers That Support and Promote Child and Youth Mental Health in Sweden—A Qualitative Interview Study

**DOI:** 10.3390/ijerph19137730

**Published:** 2022-06-23

**Authors:** Maria Fjellfeldt

**Affiliations:** Department of Health and Welfare, Dalarna University, 791 31 Falun, Sweden; mfj@du.se; Tel.: +46-23-778-494

**Keywords:** sustainability, collaboration, child and youth mental health, welfare providers, social services

## Abstract

When addressing child and youth mental health, policy makers around the world call for collaboration between welfare providers. Research shows, however, that cross-sector collaboration is challenging. This article aims to scrutinize the issue of sustainability in the collaborative work undertaken between welfare providers to jointly support and promote child and youth mental health. In a qualitative interview study, 19 key officials involved in collaborative mental health work in three Swedish municipalities were interviewed, 13 individually and 6 in three small groups. Data were analyzed through content analysis and the application of practice-oriented collaboration theories. The results show that informants feel collaboration is beneficial for child and youth mental health. The results also show that five aspects of this collaborative work can affect its sustainability: (1) how the collaborative work was set up: if it was a special project or part of existing organizational structures; (2) what model of funding was used; (3) how many organizational levels were involved; (4) if goals were common, concurrent or contradictive; and (5) if important stakeholders were seen to be ‘missing’. Collaboration members felt their collaborative work had caused them to drift away from important non-participant stakeholders. This article concludes that to develop long-term sustainable collaborations addressing child and youth mental health, key features of collaborative work need to be taken into consideration.

## 1. Introduction

In order to address child and youth mental health issues, policy makers routinely call for collaboration between welfare providers [1,2,3]. In Sweden, as in other countries, policy concerning child and youth mental health has evolved during the last decade [4,5,6]. Since 2015, one of the main policy objectives has been to increase collaboration between welfare providers [7].

When collaborating, group members usually seek “to achieve some end that they could not have achieved independently” [8] (p. 240). However, making collaboration work in practice is often described as a slow process that takes both time and resources. The combination of high expectations and organizational inertia often results in disappointment. The concept ‘collaborative inertia’ [9] has emerged to describe the fact that collaborative arrangements often appear to have a negligible or extremely slow rate of output. Implementing collaboration has often been described as more difficult than anybody could have imagined [1]; good interprofessional collaboration is often experienced as demanding to achieve [10]. Some research even suggests that some problems are best solved by individual organizations working alone [11]. Collaboration is recommended only when there is no other choice [12]. Some social problems, however, cannot be easily solved except by working across sector lines to develop shared understandings and a commitment to shared solutions [11,13]. Countering mental illness among children and young people has been identified as just one such problem [10,14]. For collaborations to work in contexts such as this, it is argued that they need to be multifaceted, dynamic, and functional across many different levels [13].

Barriers to successful collaboration between organizations addressing child and youth mental health have been identified as: (1) a lack of trust between group members; (2) problems securing long-term financing; and (3) achieving a shared sense of commitment and responsibility among leaders of the participating organizations at the highest level [1,15,16,17].

In the literature on collaboration, a tension between the principles of ‘specialist knowledge’ and ‘holistic knowledge’ has been described [18]. During the 20th century, the prevailing practice expected of specific professional groups was ‘specialist knowledge’. In the 21st century, the ‘holistic knowledge’ acquired through more collaborative and team-based practice has been increasingly advocated [18]. In the Swedish welfare context, for example, implementing ‘holistic governance’ is seen to take place through a process of policy integration. Government strategists looking to address complex social problems with the principle of ‘holistic knowledge’ in mind frequently advocate collaboration between different welfare organizations [19].

Two useful frameworks have emerged from recent studies examining collaborations which have successfully addressed complex social issues: the collective impact [11] and the conceptual framework [20], both derived from research in different disciplines and geographical contexts. Collective impact is defined as ‘the commitment of a group of important actors from different sectors to a common agenda for solving a specific social problem.’ [11] (p. 36). The claim of collective impact is that achieving large-scale social change by addressing complex social issues comes from better cross-sector coordination rather than from the isolated interventions of individual organizations. Thus, isolated impact is understood as an approach oriented towards finding and funding a solution embodied within a single organization. Kania and Kramer question [11] whether isolated impact is the best way to solve complex social problems in the contemporary interdependent world. The shift from isolated impact to collective impact, however, requires a systematic approach that focuses on the relationship between organizations and their progress toward shared objectives. The conceptual framework [20] is based on extant literature of successfully integrated health and social care and defines enabling integration factors that support collaborative integration efforts between healthcare and social services organizations. Taken together, these two frameworks identify three factors as of crucial importance in successful collaborations: (1) a shared/common vision/agenda and a clear set of goals, (2) continuous communication, and (3) shared measurement and accountability agreements. Collective impact [11] includes two additional success factors: (4) mutually reinforcing activities, meaning that stakeholders undertake activities that support the actions of others and work in coordination with them, and (5) backbone support organizations, meaning that a separate organization should be implemented that can plan, manage, and support the initiative’s collective impact. The conceptual framework also adds effective leadership, a collaborative model founded on team-based care and dedicated resources and financing as crucial elements in a successful collaboration. 

As this overview has shown, existing research has characterized collaborative work between welfare providers as a significant challenge. At the same time, collaboration has sometimes been deemed necessary, if not beneficial, in order to jointly address complex issues. However, there is a lack of research exploring aspects of sustainability in the collaborative efforts addressing child and youth mental health. Therefore, the aim of this article is to scrutinize aspects of sustainability in the collaborative work carried out between welfare providers as they jointly strive to support and promote child and youth mental health. The research questions this study posed were: what are the characteristics of sustainable collaborative practices addressing child and youth mental health? How can selected examples of empirical collaborations in practice be understood from a perspective of developing long-term sustainable collaborative work addressing child and youth mental health?

This study is part of a larger project that addresses the process of implementing a broadened mental health policy in Sweden at a national, regional, and local level [14,21,22,23]. This part of the project focuses on the issue of sustainability in the collaborative work undertaken between welfare providers to jointly support and promote child and youth mental health. The intention is that by examining one country’s experiences (in this case Sweden), an improved approach might be identified for other countries struggling with similar concerns.

## 2. Analytical Framework

To deepen the understanding of the empirical data collected for this study, an analytical framework that follows a practice-oriented theory on collaboration will be adopted. The empirical data will be scrutinized using the following four dimensions: (1) organizational structures, (2) funding models, (3) goals in collaboration, and (4) membership structures.

The first dimension will address the question of how the collaborative work was implemented in relation to existing organizational structures. The second dimension will explore how the collaboration was financed. In the Swedish context, collaborations between welfare providers have often been organized as separate projects with their own identities that run in parallel with existing organizational structures. These projects are often directly funded by external agencies and their financing is usually situated outside of ordinary organizational budgets. When studying the effectiveness of this strategy [24], it was found that the favorable conditions for collaboration established during the project’s duration were not sustained after the project had ended.

The third and fourth dimensions that will be applied to the empirical data have been taken from Huxham [12], who has sought to develop a theoretical model for practical collaboration. Two of the crucial aspects of collaboration that were identified are common goals and membership structures. Huxham claims that it is necessary to have a clear set of goals if the partners are to work together effectively to operate shared policies. 

Goals can be explicit, assumed by one partner but not recognized by another, or deliberately hidden. Goals can also be common, organizational or personal. It can be a challenge to reach agreement on goal formulations because these are often communicated and negotiated between stakeholders positioned in different professional and organizational cultures that employ different terminology and logics.

The final dimension relates to the question of Huxham [12] who takes part in a collaborative effort. Sometimes the membership structure of a collaboration is not clear or lacks formal membership documentation. It can also be difficult to know on which organizational level partners in a collaboration are operating. Collaboration can also be affected by organizational changes, such as in cases when group members are not present or are replaced. It can be difficult to agree on common goals, build trust and handle power when it is unclear who is a member in the collaboration and who is not. The way the different members approach their mandate affects what is and is not possible in the collaboration.

## 3. Materials and Methods

In order to scrutinize aspects of sustainability in the collaborative work to support and promote child and youth mental health undertaken between welfare providers, a qualitative design was considered appropriate. Qualitative interviews [25] rich in detailed information were conducted with key stakeholders. The qualitative method facilitated the emergence of nuanced in-depth knowledge about this complex research field.

The context in which this study takes place is the Swedish welfare system, where the management of education, social services and healthcare is divided across two state organizations (municipalities and county councils) with their own individual structures and decision-making processes. Municipalities are responsible for the organization of schools and citizens’ social welfare services, whereas county councils are responsible for managing primary and specialized healthcare.

To kick off the implementation of a new policy process in 2015 [7], more than EUR 84,000,000 was allocated to establish the conditions in which municipalities and county councils could develop joint long-term operations in the field of mental health.

Based on evidence gathered from previous studies [21,22], three collaborations in three different municipalities were strategically selected (see Table 1). Each collaboration was selected to represent a diversity in the form of collaboration, its geographical location and the population of the municipality in which it was located. Each collaboration was also chosen because it represents a collaborative form that frequently recurs in the Swedish welfare context [21,22]. Therefore, the collaborations examined in this article can be assumed to be representative trying to understand contemporary professional collaborative practice. The following three collaborative forms were included in this study:First-line mental health support in a small municipality in the southern part of Sweden. Children and young people (aged 6–18 years) and their parents could access support for mild or moderate mental health issues. The service was intended to help identify and treat children and young people’s mental health needs at an early stage, in order to prevent more severe conditions from developing. The collaboration included representatives from social services and primary and specialized psychiatric care.Consultation team in a rural municipality in the middle part of Sweden. A group of about ten professionals with responsibility for the mental health of children at school (aged 7–16) discussed, on a monthly basis, cases where one of the members felt the mental health of a pupil was at risk. The cases were discussed anonymously. Members discussed what needed to be implemented and who would do it. The collaboration included individuals representing local schools, social services, and primary care agencies.Children’s house in a large municipality in the middle part of Sweden. The professionals in this collaboration coordinated activities around children and young people (0–17 years) who were victims of domestic violence or sexual abuse (regardless of their relationship with the perpetrator). The collaboration included members of the social services, pediatrics and psychiatry specialized care units, police, prosecutor’s office, and county administrative board. When investigating the children’s situation, a child visited one particular house, where they could meet all various professionals involved in their specific case. The child did not have to pay one visit at the police station, one visit at the social service office, one visit at the pediatric specialized care, etc.

For each collaboration, the approach considered most appropriate to gain a nuanced and in-depth knowledge of the issues was snowball sampling [26]. Key people within each collaboration, that is the coordinators, were identified and contacted by email. Coordinators were asked to provide the names of individuals involved in the collaborations who might be willing to take part in an interview. These people were then contacted by mail and telephone, received written and oral information about the study, and were asked to participate. In the first-line mental health support 4 informants were recruited, in the consultation team 5 informants were recruited, and in the children’s house 10 informants were recruited. In all, 19 persons agreed to participate and gave their oral and written consent. Semi-structured interviews [27] were conducted between September 2019 and April 2021. In total, 15 interviews were conducted: 12 individual interviews and 3 group interviews. In 2019, all five interviews were conducted face to face. In 2021, the remaining ten interviews were conducted as online video meetings due to COVID-19 restrictions. The interviews included questions such as: How would you describe the collaborative work that you are a part of? Why are you collaborating? Who initiated this collaboration and why? What is your and your organization’s role in this collaboration? Are there any difficulties you have encountered with the collaborating process? Table 1 provides a full overview of the interviews that were carried out as part of this study.

Individual interviews, which lasted between 30–75 min, were intended to give informants the opportunity to develop their answers at their own pace without being influenced or interrupted by others. In the group interviews, however, which lasted about 90 min, the potential for dynamic interactions between individuals was utilized to facilitate more spontaneous expressions and perceptions [27]. The three group interviews were suggested to be held by the informants themselves. In two of the group interviews, the informants comprised individuals working in specialized psychiatry. In the third group, the informants were coordinators working in social services. Informants who suggested group interviews worked closely together and were positioned at the same organizational level. No informant was in a position of dependence on another. This meant they could speak openly during the interview without worrying about undesirable consequences. During the group interviews, the researcher simply asked each question and then paused so informants could answer. Sometimes one informant answered and the other affirmed the content by nods or positive verbal expressions. Sometimes informants took turns when answering and reasoned together.

In order to facilitate participation in the study, one informant from the large municipality (representing the police) provided written answers and one informant (representing the prosecutor’s office) was interviewed by telephone. The interview data were recorded and transcribed verbatim.

During the analysis, transcribed interviews were read multiple times. The data were sorted initially using an inductive approach according to conventional content analysis [28]. Core categories were identified and defined. Secondly, to allow for a deeper understanding of the empirical data to emerge, the data were related to selected concepts identified in collaboration theory [12,18,24] in a directed content analysis [28].

External validation was addressed when the preliminary results were critically reviewed at a national conference. Five researchers representing the fields of social work, pedagogy and mental health commented on the analysis of results. Elaborating on certain themes was suggested and the analysis was further developed as a result. The results from this more refined analysis process were then presented at a professional seminar at Dalarna University, where three senior researchers in health and welfare critically reviewed the adjusted manuscript. This review resulted in still further development of the manuscript’s approach. Implementation theories were excluded and a particular focus on aspects of sustainability was expanded.

## 4. Results

### 4.1. Organizational Structure, Funding Models and Organizational Levels Involved

As the interviews showed, how the collaborations were set up and how they were financed differed. In the first-line collaboration, informants had been seconded for a time-limited period to work on a project that had been set up in parallel with the organizations they each worked for. Their contracts had been financed by special government funds that aimed to stimulate local collaborative work in the mental health field. At the time of the interviews, the original two-year period had just been extended for another six months.


*It [the project] is temporary. Right now, in March, they are working on making a new decision. They have said that we can continue for the rest of the year. That is the information we have right now (Group interview, first-line collaboration, psychologist in primary care).*


This time limitation meant that the collaboration was only formalized in the organization in the short term, without any thought for its long-term sustainability. The uncertain future of the work was a cause for concern among the collaboration’s members. The uncertainty of the future affected the mental health counselors. It was hard to go ‘all in’ when there was no long-term perspective guiding the collaboration.

The question of continuing the collaboration or not was perceived by the informants only as an economic issue. No one believed that someone in a decision-making position really wanted to close down what had become a greatly valued and much used collaboration. The informants valued the direct government funding since it made it possible to develop the collaboration. However, they were also critical of this type of funding because the work that had been built-up around it could not be easily integrated back into their primary organizations. The organizational priority of ‘greatest need’ would make it difficult to allocate money for the sort of preventive work first line was doing from ordinary budgets. First-line’s mental health support was thus only a temporary fix. It lacked long-term sustainability as a dominant perspective.

In the consultation collaboration, the organizational and financial situation were somewhat different. In this municipality, collaborative work addressing child and youth mental health had previously been organized as a project. Direct government funds had made it possible for the secondment of one social service official with the task of establishing a collaboration addressing child and youth mental health. However, this person resigned after six months. One informant described how this person had experienced her role:


*She said, ‘I cannot bear to run this anymore’, so she was happy when we said we could look at this and make it work better (Individual interview, consultation collaboration, quality strategist in the authority’s education department).*


Another informant in the consultation collaboration described the direct funding allocated to establish the conditions in which municipalities and county councils could develop joint long-term operations in the field of mental health as an insufficient and short-term solution:


*It is a bit of a disadvantage, with a little bit of stimulant funding, you can get a project that really shines. But then you can hardly afford to introduce it in the ordinary organization. However, this project manager became an important person, because she focused on getting this work done (Individual interview, consultation collaboration, department manager in social services).*


The consultation collaboration was a result of some restructuring that had taken place when a short-term collaboration project had ended. Instead, a long-term perspective had been implemented and the collaboration had now been integrated into the partners’ regular organizational workflow. However, the officials involved in the collaboration did not have their job descriptions adjusted so they had adequate time to carry out this additional task. Instead, they had to make it fit into their ordinary budgets and duties. Previous collaborative experiences meant that subsequent collaborations were shaped according to a different collaborative form. Rather than being run by a single individual on a time-limited project, managers, politicians, and civil service principals were now part of the implementation process. The collaboration was no longer a project but had become an integral part of the work carried out by each of the partner organizations.

In the children’s house collaboration, the collaborative work was embedded in the ordinary budgets and structures of each of the participating organizations. One informant thought that the collaboration had initially been financed at least partly by direct government funds, but no one knew for certain because this was more than twenty years ago. At the time the interviews took place there was no external funding financing this collaboration.

Each of the collaborations also differed in terms of the organizational levels at which they operated. The results showed that the first-line collaboration was conducted mostly by operational and other front-line staff. With the consultation collaboration, the results showed that in addition to operational staff, staff from the political, management, and strategic levels were also involved, at least amongst the education and social services partners. Primary healthcare services, however, continued to be represented only on a front-line level. For the children’s house collaboration, most organizational levels of all of the member organizations were involved in various ways. A *steering group* consisting of senior managers anchored the collaboration in regular operations through organizational budgets, staffing, premises, and writing contracts. A *strategic group* consisting of persons in managerial positions or possessing expert knowledge planned the routines and general practices. *Operative groups* consisting of front-line staff collaborated around specific cases. There was also a *reference group* where front-line staff or other persons interested but not directly involved in the collaboration could discuss issues of relevance.

### 4.2. Common, Concurrent, and Contradictive Goals

In all of the collaborative forms examined in this study, one common goal was to bring together the full range of professional perspectives on the situation facing each child or young person so that a more holistic view could be taken.


*This consultation team is a way to try to see more of the whole in the child and in the family (Individual interview, consultation collaboration, department manager in social services).*


A holistic view can be understood as a way of achieving a goal that no one could have achieved independently [8] (Provan and Kenis, 2008). However, informants occasionally expressed that they had additional organizational goals from those articulated by the collaboration. In the interviews with the consultation collaboration, for example, informants from the education authority described how they hoped the collaboration would lead the children and young people to achieve better school results, while informants from the social services articulated their desire for the decreased effects of mental illness among young people. These partners thus displayed two additional and different organizational goals. A similar diversity of organizational goals was also mentioned by one informant in the children’s house collaboration:


*We have different goals and points of departure in these matters where we work together. The police should investigate the crime, we will do another thing and so on. I think it’s still good to have that forum to highlight each other’s different perspectives and that you sort of find ways forward (Individual interview, children’s house, group leader at the reception unit for children and families in social services).*


Informants described how partners in the collaboration had one common goal—a holistic approach informed by different perspectives—but that they also had organizational goals which they hoped the collaboration would help them to achieve. In the two cases cited here, the additional organizational goals were in line with the common goal and could therefore be achieved in parallel with each other.

In the first-line collaboration, two organizational goals were described as present at the management level. One was to identify and meet mental health needs at an early stage to prevent the development of more severe conditions that require more extensive treatment. The other was to save money. Informants explained however that the economic goal of saving money by preventing severe mental health conditions was not fulfilled in practice:


*So they [politicians] had a hypothesis that this would save money as well. And it turned out that no, you certainly do not save any money. More [people] are coming (Group interview, first-line collaboration, counsellor and family therapist in primary care).*


What had not been anticipated was that a new, previously invisible group would start to use the first-line service. Network-weak families who needed advice and support saw it as less stigmatizing to contact than the specialist psychiatry service, which placed unexpected financial demands on services initially set up to address child and youth mental health.

This meant that in the first-line collaboration there were two contradictory organizational goals, neither of which could be achieved at the same time. A tension arose between the ambition to meet mental health needs at an early stage or to save money.

### 4.3. Membership Structures

Informants’ experiences of their membership in their collaborations varied. In the children’s house collaboration, membership was a long-term commitment agreed on in written contracts. Membership was written into the job descriptions of those who were involved with it. When one member quit, someone else took over their position. None of the informants interviewed here had been involved in the collaboration from its beginning. One informant said: *The collaboration was already established when I came to this position (Individual interview, children’s house, expert in domestic violence at the county administrative board).* None of the informants felt that any specific organization was missing from this particular partnership. In fact, a representative from an organization currently outside the collaboration, the county administrative board, had recently joined the group so that they could contribute their expertise regarding education and training in domestic violence issues.

In the first-line and consultation collaborations, the membership structures were described differently. Informants from both of these collaborations felt a vital stakeholder was missing. In the first-line collaboration, informants felt that their holistic approach could be improved through the presence of someone from the education authority. In the consultation collaboration, informants felt that the specialist psychiatry service ought to be included. All of the young people the collaboration had dealt with so far could have been treated more holistically if there had been a specialist psychiatry perspective in the group. The absence of the specialist psychiatry service, despite requests from the consultation group for their inclusion, meant that it was drifting farther away from the collaboration members. This had caused frustration among members within the collaboration:


*The more they [child and adolescent psychiatry] are absent, the more it also creates frustration unnecessarily. Because you paint pictures of “they do not do this, and they do not do that”. And then we end up even further apart from each other (Individual interview, consultation collaboration, child and youth psychotherapist in primary care).*


Informants felt that a clear sense of ‘us’ and ‘them’ had been established. They felt that the reluctance to include the psychiatry services was simply a question of resources.

In a final point about membership structures, it was found that in the first-line collaboration, all of the members understood their involvement was only ever going to be a short-term commitment.

### 4.4. A Sustainability Continuum

The three empirical examples will now be placed on a continuum measuring sustainability based on the conclusions drawn from the three different collaborative forms (Table 2). On this continuum, the first-line collaboration illustrates a ‘temporary form of collaboration’, the consultation collaboration illustrates a ‘collaboration in transition’, and the children’s house collaboration illustrates a ‘long-term sustainable collaboration’.

This ‘sustainability continuum’ makes it possible to identify the characteristics of a collaboration and extent to which it will be (or will not be) sustainable. The “guidance map” it provides will enable an analysis to be made about which aspects should be focused on if more sustainable long-term collaborations are to be developed.

## 5. Discussion

In this study, five important conclusions concerning the development of long-term sustainability in collaborations between welfare providers to support and promote child and youth mental health were identified. In this section these findings will be discussed.

The findings in this study correlate to a large extent with the factors included in the conceptual frameworks compiled by Kania and Kramer (2011) [11] and Cheng and Catallo (2020) [20]. The continuum presented in this article does not contradict these conceptual frameworks, but rather emphasizes the relevance of certain specific factors when addressing sustainability in collaborative work. These factors include shared goals, dedicated resources and financing, and funding models. This article suggests two additional dimensions are crucial when scrutinizing the sustainability of collaborative work: organizational structures and membership structures.

Firstly, the findings related to public policies, dedicated resources and financing, funding models, and organizational structures will be discussed. The results from this study showed that the collaboration form with the *closest connection* to national policy was *the least long-term sustainable collaborative form* (first line collaboration). This collaboration was described as a direct local response to national policy [7]. It was set up in parallel with the ordinary social services and healthcare organizations. Its collaborative work had a very uncertain future and was characterized by a short-term perspective both in terms of financing and organizational form. The security of the funding [1] was low. In contrast, the collaboration that was *not at all related* to national policy was the *most long-term sustainable collaboration* (children’s house collaboration). This collaboration was solidly anchored at crucial levels in all of the partner organizations. It was integrated within their ordinary organizations and future collaborative work between them was secure. It was integrated both in terms of financing and organizational form. The security of the funding [1] was stable.

These findings can be understood from a time perspective, where time has been found to be an important factor for achieving good interprofessional collaboration [1,10]. In Sweden, there are parallel lines present concerning a time perspective at the national policy level. On the one hand, one-year funding has been the standard strategy in the mental health field since 2016 [7]. On the other hand, a national investigation has recently called for a long-term perspective of at least 10 years [29]. Results from this study show that the current funding model contradicts the national ambition of implementing long-term sustainable collaborations [7]. Long-term sustainable collaborations seem to require long-term funding models, as the example of the children’s house has shown. A consequence of a short-term funding model associated with public policy seems to be short-term collaborations with low sustainability, as identified in the first-line collaboration.

The children’s house collaboration can in addition be understood as an example of a collective impact initiative [11]. The children’s house collaboration involves a centralized infrastructure, dedicated staff, a structured process, continuous communication, and mutually reinforcing activities among all participants. The collective impact character of the collaboration can thus be understood as a key to the long-term sustainability of the collaboration. Funders of the children’s house collaboration (the welfare service organizations involved) have supported a long-term process of social change and have had the patience to remain part of the collaboration for years. The understanding these stakeholders have of their role as funders is in line with the description of how funders of a collective impact initiative understand their role and commission. In contrast, the stakeholders funding the first-line collaboration hoped that the temporary funding of the collaboration would solve the economic issues confronting the individual organizations. Even though the first-line initiative included a group of important stakeholders from different sectors who were committed to a common agenda for solving a specific problem, it was not enough to achieve collective impact. Several crucial conditions were not met, not least the absence of a long-term financial perspective on the part of the funders. The first-line collaboration, however, cannot by extension be characterized as an isolated impact approach. The shared efforts described by the informants can be understood as trying to bring about a situation of collective impact, where common goals, continuous communication, and mutually reinforcing activities are desired. However, unfavorable economic conditions at the regional and local level made the development towards collective impact impossible to sustain.

Secondly, findings related to common and contradictive goals will be discussed. One goal of the first-line collaboration that completely correlated with a primary national policy objective [7] was identifying and supporting young people with mental health issues at an early stage to prevent more severe conditions from developing. Informants felt the collaboration achieved this goal to some extent when a whole new target group started to be supported. At the same time, a harsh economic reality meant that the goal of saving money by establishing early support was not reached. Financial reasons, therefore, meant that the first-line collaboration was at best only a temporary arrangement. The consensus among the research theorizing collaboration [12] and conceptualizing successful collaboration [11,20] stresses the indisputable importance of agreeing on common goals and objectives in collaborative work. Questions arising from this case concern how long-term sustainability can be achieved when goals are incompatible. How can early intervention be implemented when the collaborative partners must give priority to people with the greatest need? Do national policy makers take organizational priorities into account when developing a holistic mental health policy? How can these dilemmas be resolved, when local practices and national policy goals contradict the legislated priority of work? Could long-term sustainability in collaborative work be addressed in such cases?

Thirdly, membership structures will now be discussed from the perspective of both specialist and holistic knowledge [18,19]. The first-line and consultation collaborations can each be understood as a reaction to the use of policy integration as a government tool to implement ‘holistic governance’. Both collaborations were initially financed by direct funding associated with the idea that the implementation of collaborative work would prevent mental illness and promote mental health among children and young people. Government strategists were in employed in both of these empirical cases, just as Svensson (2022) [19] has described earlier, to promote the principle of ‘holistic knowledge’ and to advocate solutions that represented a holistic view when implementing collaboration between organizations to counter complex issues.

In the current study, the members involved in all three collaborative forms can be assumed to share an understanding of holistic knowledge as a dominant principle. However, the organizations who also have a stake in child and youth mental health but currently are not part of an existing collaboration—the education authority in one collaboration and the specialist psychiatry service in the other—are under no such obligation. These “non-participants” are likely working instead from the principle of specialist knowledge. It may be the dominant principle in their organizations and probably forms the basis for their actions. Collaboration members and non-participants can therefore be understood as representing competing principles. The tension which some members described where they felt their collaboration was causing them to drift away from non-participants can be seen as a result of these two principles at work. Members and non-participants had different opinions about what needed to be prioritized when it came to promoting child and youth mental health: the ‘whole’ situation where the understanding of holistic knowledge prevailed, or the ‘specific’ situation where specialist knowledge prevailed. The non-participant stakeholder who later joined the children’s house collaboration, for example, can be assumed to have left their ’specialist’ perspective behind and to have now adopted the members’ view of holistic knowledge as a guiding principle.

Membership structures can also be discussed from the perspective of collective or isolated impact initiatives [11]. As already mentioned, the children’s house collaboration can be understood as an example of a collective impact initiative, defined as ‘the commitment of a group of important actors from different sectors to a common agenda for solving a specific social problem.’ [11] (p. 36). All of the conditions for a collective impact are present. Since collective impact initiatives are characterized as successful collaborations, the children’s house collaboration can be understood as an attractive collaboration to join, a reputation confirmed by external actors. The other two collaborative forms—first-line collaboration and consultation collaboration—are not characterized by the typical conditions required for a collective impact initiative. These two collaborative forms can thus be understood as less attractive to join.

The implications of this study’s findings for policy, practice, and future research will now be described. Identifying the crucial elements of sustainability in a collaboration can provide a supportive tool for practitioners when trying to develop a long-term sustainable collaborative work. The continuum developed in this article could guide the evaluation of existing and future collaborations with questions such as: Where does our collaboration sit on the continuum? What do we need to think about if we want to promote our collaboration’s long-term sustainability? What can we expect from this collaborative work, given certain inflexible preconditions? Furthermore, by raising an awareness of the dilemmas that emerge when policy makers formulate goals that contradict legislated organizational priorities, this article makes it clear that a resolution needs to be sought between these rival goals or else the long-term sustainability of the collaborative work at hand will be affected. The holistic demands of early-stage prevention must be reconciled with the specialist requirements of greatest needs first.

Finally, the limitations and strengths of this study will be addressed. One limitation concerns the method design, where three collaborative forms were selected. More collaborative types could have provided additional evidence related to the research questions. However, the three collaborative forms examined here provided ‘thick descriptions’ [30] with nuanced in-depth data. This allowed for a deeper understanding of the results to evolve. Another limitation also associated with the selection procedure, concerns the snowball approach to the sampling of interviewees [26]. During the interviews with members of two of the collaborations, non-participant stakeholders were also mentioned. Representatives of non-participant organizations could have been interviewed to deepen the understanding of the collaboration’s membership structures from their perspective. To handle this limitation, the principles of specialist and holistic knowledge were included, not in the results section, but in the discussion of those results. The last limitation concerns the theories chosen to analyze the data. Other or additional theories could have been chosen, for instance, the multiple institutional logics or tangible and intangible outcomes suggested by Bryson et al. (2015) [13]. However, the theories used in this article were assessed as guiding the analysis in a way that promoted the best understanding of data.

## 6. Conclusions

This study shows that the identification of key features of a collaboration can offer an increased understanding of sustainability in collaborative work. The collaborations examined here illustrated the way sustainability interacted with three different forms of collaborative work: ‘temporary collaborations, ‘collaboration in transition’ to ‘sustainable’ and ‘long-term sustainable collaborations.’ To develop long-term sustainable collaborative work that addresses child and youth mental health, the findings from this article suggest it will be vital to: (1) integrate the collaborative work into ordinary organizational structures; (2) secure long-term funding (in ordinary budgets); (3) involve all levels of the member organizations; (4) agree on shared goals and ensure organizational and shared goals are compatible; and (5) involve all stakeholders as collaboration members. In addition, the long-term sustainable collaboration in this study was an example of a collective impact initiative. These findings will be of relevance for policy makers, managers, and practitioners, and will enable the scrutinization of an existing collaboration so that it can best support and promote child and youth mental health.

## Figures and Tables

**Table 1 ijerph-19-07730-t001:** Overview of the form and features of the collaborations and interviews conducted during this study, 2019–2021.

Collaborative Form	First-Line Mental Health Support	Consultation Team	Children’s House
Municipality character	Small municipality	Rural municipality	Large municipality
Location	South part of Sweden	Middle part of Sweden	Middle part of Sweden
Organizations represented	Social services, primary care, specialized care (psychiatry)	Social services, school, primary care	Social services, specialized care (pediatric, psychiatry), police, prosecutor’s office, county administrative board
Informants	4	5	10
Interviews	2 video individual interviews (social services, primary care) 1 video group interview (including 2 informants in specialized psychiatry)	5 individual interviews face to face	5 video and telephone individual interviews (social services, specialized pediatric care, prosecutor’s office, county administrative board) 2 video group interviews (one including 2 coordinators belonging to the social services, and one including 2 informants belonging to the specialized psychiatry care) 1 written answer (police)
Year	2021	2019	2021

**Table 2 ijerph-19-07730-t002:** A continuum of sustainability in collaborative work.

Collaboration Form	Temporary Form of Collaboration	Collaboration in Transition	Long-Term Sustainable Collaboration
Empirical example	First-line collaboration	Consultation collaboration	Children’s house collaboration
Level of sustainability	Short-term sustainability	Sustainability in transition	Long-term sustainability
	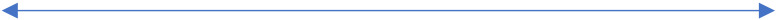 LOW HIGH
Organizational structure	Time-limited project	Former time-limited project now integrated to some extent in all members’ ordinary organizational structures	Integrated in all members’ ordinary organizational structures
Funding	Short-term external funding	Financed to some extent by all members’ ordinary budgets	Long-term financing by all members’ ordinary budgets
Organizational levels involved	Few levels involved	Involvement of organizational level varies	Most levels involved
Goals	Common and contradictive goals	Common and concurrent goals	Common and concurrent goals
Membership structures	Stakeholder experienced as missing	Stakeholder experienced as missing	Additional stakeholder had joined

## Data Availability

The author can provide all of the original data for review.

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
