# Peer review of "Developing Long-Term Sustainable Collaborations between Welfare Providers That Support and Promote Child and Youth Mental Health in Sweden—A Qualitative Interview Study"

_ijerph, 2022, doi:10.3390/ijerph19137730_

Round 1

Reviewer 1 Report

Thank you for the opportunity to review this important piece of work. It is well written and very important. I do find some similarities between the conceptual framework presented in the paper by Cheng and Catallo, 2020 and the Collective impact framework first outlined by Kanier and Kamer, 2011.Kania, J., Kramer, M. (2011). Collective impact. Stanford Social Innovation Review, 9(1), 31–35.

In fact the constructs are almost exactly the same. I would like to see some inclusion of this framework in the conceptualisation and the discussion or some justification in the paper as to why this framework is not appropriate for this study. This framework may also provide additional richness for the discussion in this paper. To improve the rigor and relevance of the paper, I recommend looking at what has been done in the Collective Impact literature and including some comparisons in this paper. How can you expand on this to strengthen these types of collaborations (which are extremely important). 

Author Response

Dear Reviewer 1,

Thank you so much for constructive comments. They have helped me to improve the manuscript as follows:

According to you the research design, the description of methods, the presentation of results and the conclusions could be improved. These parts of the manuscript have been improved.

According to the excellent suggestion the Collective impact framework (Kania & Kramer, 2011) was included throughout the manuscript.

Thank you for reviewing this manuscript.

Kind regards,

Maria

Reviewer 2 Report

Title: please specify place and methods

Line 29-30: is it needed? Justify or delete

Methods: You investigated collaborations between welfare providers that support and promote child mental health. Please update your title and objectives accordingly.

Please clarify number of interviews, number of individual interviews, number of group interviews in abstract and methods. How did you conduct group interviews? Provide characteristics of the groups?

Please provide interview guidelines used for individual and group interviews. You may provide these in a box or table.

Results: Please review your quotations and provide more appropriate representative illustrative quotations. For each quotation, please specify the type of interview, type of collaboration and respondent’s characteristics.

Discussion: Majority of your discussion is not supported by evidence. Please cite appropriate reference to support your arguments in discussion.

Avoid using future tense. You have already conducted this study and written this manuscript.

Author Response

Dear Reviewer 2,

Thank you so much for constructive comments. They have helped me to improve the manuscript as follows:

  • In the title place and methods were specified.
  • Line 29-30 were deleted. This information was outside the scope of the article.
  • Suggestion: “You investigated collaborations between welfare providers that support and promote child mental health. Please update your title and objectives accordingly.”

My response to the suggestion: I have investigated collaborations between welfare providers that support and promote youth and child mental health. The title and objectives, and the manuscript as a whole, have been updated accordingly.

  • The number of interviews, individual interviews and group interviews were clarified in abstract and methods.
  • Characteristics of the groups were provided in the methods.
  • The group interviews were described in detail in the methods. I hope they are not described in too much detail.
  • Questions used during the interviews were added to the methods. I preferred to include the questions in the text rather than providing the whole list of questions in the guidelines. Of course, I could also provide the full interview guidelines if needed, but I thought they would be hard to put in here when I refer to table 1 repeatedly in this part method section, and the existing table is now placed when all information about form and features have been provided in text. If the interview guidelines should be included as a table, this would affect the appropriateness of placing the already existing table at its current place. I therefore decided to include examples of interview questions in the text.
  • In the result section the type of interview, type of collaboration and respondent’s characteristics was specified for each quotation.
  • Quotations were reviewed. In one quotation two sentences were added to better illustrate the whole picture of the content of the result presented. However, no quotations were exchanged as suggested by reviewer 2. The decision not to exchange quotations were based on lack of comments on this from reviewer 1, and the Reviewer 1 comment that the manuscript was “well written and very important.”
  • The discussion and conclusion were improved by including the Collective impact framework in the text. This framework constitutes additional evidence asked for by the reviewer 2 to support the arguments of the discussion.

Thank you so much for reviewing my manuscript.

Kind regards,

Maria